# Molecular Identification of *Culicoides* Species and Host Preference Blood Meal in the African Horse Sickness Outbreak-Affected Area in Hua Hin District, Prachuap Khiri Khan Province, Thailand

**DOI:** 10.3390/insects14040369

**Published:** 2023-04-08

**Authors:** Ketsarin Kamyingkird, Suchada Choocherd, Wissanuwat Chimnoi, Nutsuda Klinkaew, Chanya Kengradomkij, Pornkamol Phoosangwalthong, Nipa Thammasonthijarern, Khampee Pattanatanang, Tawin Inpankaew, Jumnongjit Phasuk, Burin Nimsuphan

**Affiliations:** Department of Parasitology, Faculty of Veterinary Medicine, Kasetsart University, Lad Yao, Chatuchak, Bangkok 10900, Thailand; ketsarinkamy@hotmail.com (K.K.);

**Keywords:** *Culicoides*, African horse sickness, blood meal analysis, host preference blood feeding

## Abstract

**Simple Summary:**

This study is to investigate the potential vectors of African horse sickness (AHS). *Culicoides* were collected near horse stables in Hua Hin district, Prachuab Khiri Khan province, Thailand, where horses were affected and died from AHS in 2020. Twelve *Culicoides* species were identified. The predominant *Culicoides* species in all farms was *C. oxystoma* followed by *C. imicola.* The *Culicoides* collected in this study fed on blood from horses, dogs, pigs, and humans for their blood meal. This study has identified the potential AHS vector *Culicoides* species and its zoonotic potential in this area for the first time.

**Abstract:**

African horse sickness (AHS) was reported as an outbreak in Thailand in 2020. Hematophagous insects from the genus *Culicoides* are the suspected vector responsible for AHS transmission. Horses in Hua Hin district, Prachuab Khiri Khan province, Thailand, were affected and died from AHS in 2020. However, the potential *Culicoides* species and its host preference blood meal in the affected areas are unknown. To investigate the potential vectors of AHS, *Culicoides* were collected using ultraviolet light traps placed near horse stables. Six horse farms, including five farms with AHS history and one farm without AHS history, were included in this study. Morphological and molecular identification of the *Culicoides* species was performed. Polymerase chain reaction (PCR) targeting the cytochrome b oxidase I (COXI) gene for confirmation of the *Culicoides* species, identification of the prepronociceptin (PNOC) gene for host preference blood meal, and bidirectional sequencing were conducted. Consequently, 1008 female *Culicoides* were collected, consisting of 708 and 300 samples captured at positions A and B at a distance of <2 and >5 m from the horse, respectively. Twelve *Culicoides* species identified by morphology were noted, including *C. oxystoma* (71.92%), *C. imicola* (20.44%), *C. actoni* (2.28%), *C. flavipunctatus* (1.98%), *C. asiana* (0.99%), *C. peregrinus* (0.60%), *C. huffi* (0.60%), *C. brevitarsis* (0.40%), *C. innoxius* (0.30%), *C. histrio* (0.30%), *C. minimus* (0.10%), and *C. geminus* (0.10%). The PCR detection of the *Culicoides* COXI gene confirmed *Culicoides* species in 23 DNA samples. PCR targeting the PNOC gene revealed that the *Culicoides* collected in this study fed on *Equus caballus* (86.25%), *Canis lupus familiaris* (6.25%), *Sus scrofa* (3.75%), and *Homo sapiens* (3.75%) for their blood meal. Human blood was identified from two samples of *C. oxystoma* and a sample of *C. imicola*. Three dominant species including *C. oxystoma*, *C. imicola,* and *C. actoni* that were reported in the Hua Hin area prefer to feed on horse blood. Moreover, *C. oxystoma*, *C. imicola*, and *C. bravatarsis* also feed on canine blood. This study revealed the species of *Culicoides* in Hua Hin district, Thailand, after the AHS outbreak.

## 1. Introduction

African horse sickness (AHS) is a disease caused by *orbivirus* in the subfamily Sedoreovirinae, family Reoviridae. It is an arthropod-borne noncontagious disease of Equidae, which is endemic in tropical and subtropical areas of Africa [1], epizootic in Europe, and reported in the Middle East, India, and Asia [2]. In Africa, seasonal occurrences of AHS appear in late summer, which are related to the midge population. However, outbreaks have also been reported in Southeast Asia and the Iberian Peninsula by the spreading of the competent infected biting midge through wind and active flight [1]. In Southeast Asia, the AHS outbreak suddenly and seriously caused ~95% of the fatality rate in domesticated horses in Thailand and Malaysia from March to May 2020 [3]. At that time, AHS virus (AHSV) serotype 1 had been identified in Thailand [4,5]. Almost 600 horses died, directly impacting the country’s horse industry. Hence, the outbreak has also raised animal welfare issues as equids were kept in nets for long periods to protect them from vectors [6]. The most relevant competent vectors of AHSV in Africa are *Culicoides (C.) imicola* and *C. bolitinos* [1]. However, the vector competence for AHSV in other continents such as America, Australia, and Southeast Asia, including Thailand, is still unknown [1]. Therefore, the identification of *Culicoides* species in the AHSV transmission would be an effective beginning to an investigation of the competence vector of AHSV.

Studies on *Culicoides* host preference patterns help to understand the epidemiology and transmission networks of vector-borne pathogens. It helps to understand the behavior of *Culicoides* and the possibility of disease transmission [7]. Thus, the molecular detection of blood meal sources on *Culicoides* was developed [8]. Two types of primers can be used for host identification of blood meal in female *Culicoides* including (1) species-specific primers, which provide fast and cheap identification, and (2) universal primers, which provide a broad host range and mixed blood meal identification [9]. Several feeding preferences of *Culicoides* studies were conducted in Europe [9] including France [10], Denmark [11], and Southeastern Serbia [1]. Mammal species, including livestock, domestic wild animals, and bird species were identified as the blood meal preferences of *Culicoides* [8,11]. Most of the study was focused on the *Culicoides* diversity [12,13,14,15], the detection of emerging parasitic pathogens [16], and leishmaniosis in humans [17], but not related to AHS.

*Culicoides* are also known as “Rin″ in Thai; they are well-known pests of humans and many animal species. *Culicoides* have been identified in animal sheds, mangroves, and beaches along the Andaman coastal region [13], as well as horse farms [18] in Thailand. In Thailand, *Culicoides* species of fauna have been investigated since 1938 and ~100 *Culicoides* species were recently recorded [19]. However, few *Culicoides* host preference studies have been conducted [7]. This study aimed to identify the species *Culicoides* and its host preference blood meal in horse farms with AHS outbreak history in Hua Hin district, Prachuab Khiri Khan province, Thailand.

## 2. Materials and Methods

### 2.1. Study Design and Data Collection

This study was designed to collect *Culicoides* in horse farms with AHS-positive history. The history of AHS-positive farms was received from the head of the Hua Hin Department of Livestock Office in Prachuap Khiri Khan province. To identify AHS-positive farms, the Department of Livestock Development (DLD) officers introduced and provided the locations of the farms to be considered for this study. Other animal species, aside from the equids, being raised in or near the horse stables were observed. Latitude, longitude, temperature, humidity, and time were recorded at each trapping site. Two traps were used on each farm.

Five farms with positive history of AHS (farms A–E) and one farm that was historically negative for AHS (farm F) were included (Figure 1). All farms were located in suburban areas of Hua Hin district. The location of each farm included farm A: 12°33′17.3″ N 99°56′46.2″ E; farm B: 12°34′34.4″ N 99°53′09.9″ E; farm C: 12°33′08.8″ N 99°56′59.3″ E; farm D: 12.546426″ N, 99.955316″ E; farm E: 12°33′12.3″ N 99°57′26.7″ E; and farm F: 12°33′00.6″ N 99°57′35.6″ E, respectively. There were 1 to 10 horses on each farm. Chickens, dogs, cats, cattle, and humans inhabit near the horse farms. The average temperature and humidity ranged from 25.5 °C to 30.8 °C and 68% to 83%, respectively.

### 2.2. Sample Collection and Identification

*Culicoides* collection: Ultraviolet (UV) light traps (purchased from John W. Hock Co. Ltd., Gainesville, FL, USA) were placed near the horse <2 m (position A) and far from the horse >5 m (position B) for 12 h (from 6 PM to 6 AM) in March 2021. *Culicoides* collected from each trap were counted. Fully fed female *Culicoides* were sorted from other insects and used for morphological and molecular identification. The use of animals and collection protocol was approved by the Institutional Animal Care and Use Committee of Kasetsart University (approval number; ACKU64-VET-015), which is in accordance with the guidelines of animal care and use under the Ethical Review Board of the Office of National Research Council of Thailand (NRCT) for the conduction of scientific research and ARRIVE (Animal Research Reporting of In Vivo Experiments) guidelines.

Morphological identification: Fully fed female *Culicoides* were dissected. Briefly, the head, thorax, wings, legs, and abdominal spermatheca were carefully dissected and placed on the glass slide and prepared in Hoyer’s media. Morphological identification of *Culicoides* species was conducted in all fully fed female *Culicoides* specimens followed by identification key as previously described [20,21,22,23].

DNA extraction: *Culicoides* abdomen containing host blood meal was individually preserved in absolute ethanol before DNA extraction. DNA extraction was conducted using DNA extraction kit (MACHEREY-NAGEL, Dueren, Germany). DNA concentration was measured using spectrophotometer (Eppendorf, Hamburg, Germany) and stored at −20 °C as DNA template.

Molecular identification of *Culicoides* species: To confirm the morphological identification results, 26 *Culicoides* DNA samples were selected for molecular identification of *Culicoides* species by PCR using the cytochrome b oxidase I (COXI) gene. Forward primer (C1J1718 [5′-GGAGGATTTGGAAATTGATTAGT-3′]) and reverse primer (C1N2191 [5′-CAGGTAAAATTAAAATATAAACTTCTGG-3′]) amplifying 830 bp of polymerase chain reaction (PCR) product were used as previously described [24]. PCR reaction was performed in 25 µL per reaction. Each PCR reaction contained 1 × PCR buffer, 1.5 mM MgCl_2_, 0.2 mM each dNTPs, 0.2 pM of forward and reverse primers, 0.02 unit of Taq polymerase (Invitrogen, Carlsbad, CA, USA), and 2.5 µL of *Culicoides* DNA template. *Culicoides* and dog DNA templates were used as positive and negative controls, respectively. Thermocycler conditions included initial denaturation at 95 °C (5 s); five cycles of 94 °C (40 s), 45 °C (40 s), and 72 °C (60 s); followed by 45 cycles of 94 °C (40 s), 50 °C (40 s), and 72 °C (1 s); and final elongation at 72 °C (7 min).

### 2.3. Blood Meal Identification

Identification of host preference blood meal of *Culicoides* was conducted using PCR targeting prepronociceptin (PNOC) gene as previously described [24]. Although the COI gene is the better target for identifying a broad range of vertebrate host blood origin in *Culicoides*, in this study, the PNOC gene was used to confirm host preference of *Culicoides* that was consumed within the last 48 h to avoid detection of old and digested blood remaining in the *Culicoides* midgut. A set of primers including PNOC forward primer (5′-GCATCCTTGAGTGTGAAGAGAA-3′) and PNOC reverse primer [5′-TGCCTCATAAACTCACTGAACC-3′], which amplify 330 bp of PCR products, were used. The PCR mixture in 10 µL reactions included 1 × PCR buffer, 0.25 mM dNTPs, 0.75 mM MgCl_2_, 0.05 pM of each primer, 0.025 unit of Taq polymerase (Invitrogen), and 2 µL of *Culicoides* DNA template extracted above. Thermocycler conditions include initial denaturation at 95 °C 5 min; 35 cycles of 95 °C (30 s), 55 °C (30 s), and 72 °C (45 s); followed by final elongation at 72 °C (5 min) [24].

### 2.4. Sequencing and Bioinformatics Analysis

PCR products were purified using a gel extraction kit (MACHEREY-NAGEL, Dueren, Germany) and used for nucleotide sequencing. Nucleotide sequence quality was monitored in chromatogram using Unipro UGENE software (version 41.0; [25]). Nucleotide sequences were compared to reference sequences using the Basic Local Alignment Search Tool in the National Center for Biotechnology Information. The phylogenetic tree was constructed using the maximum likelihood method and General Time Reversible model in MEGA11 [26].

## 3. Results

### 3.1. Culicoides Identified in Horse Farms

Fully fed female *Culicoides* collected in this study were 708 and 300 *Culicoides* from position A and B, respectively. There were 289, 125, 3, 227, 48, and 16 *Culicoides* collected at both position A and B from all farms (Table 1). Moreover, of 1008 fully fed female *Culicoides*, 12 *Culicoides* species were found in this study. *Culicoides oxystoma*, *C. imicola*, *C. actoni*, *C. flavipunctatus*, *C. asiana*, *C. peregrinus*, *C. huffi*, *C. brevitarsis*, *C. innoxius*, *C. histrio*, *C. minimus*, and *C. geminus* were identified morphologically. The predominant *Culicoides* species in all farms was *C. oxystoma* (725/1008; 71.92%) followed by *C. imicola* (206/1008; 20.44%; Table 2, Figure 2). Wings of the 12 female *Culicoides* species collected in this study were also indicated (Figure 3). 

The molecular identification and nucleotide sequencing of 23 *Culicoides* DNA samples helped to confirm the species. The molecular results were consistent with morphological identification. Sequencing analysis of COXI genes among 23 *Culicoides* samples ranged from 83.33% to 100% identity to the GenBank. Phylogenetic analysis distinguished each *Culicoides* species into different clad and was identical to the reference sequences in GenBank (Figure 4).

### 3.2. Blood Meal Identification in Fully Fed Culicoides in Hua Hin District, Prachuab Khiri Khan Province, Thailand

Eighty *Culicoides* DNA samples were confirmed for host preference blood meal identification. The molecular identification of host preference blood meal in *Culicoides* collected in this study showed that *Culicoides* prefer to feed on horse blood (*Equus caballus*; 86.25%, 69/80) as the meal (Figure 5). The blood of *Canis lupus familiaris* (3.75%, 3/80), *Sus scrofa* (3.75%, 3/80), and *Homo sapiens* (3.75%, 3/80; Figure 5) was also identified as the blood meal in the collected *Culicoides* in Hua Hin district, Prachuab Khiri Khan province, Thailand.

Horses, dogs, pigs, and human blood were fed on by ten, three, two, and two *Culicoides* species, respectively. *C. oxystoma* was found to be the predominant species feeding on all types of animal blood meals. *C. imicola* was also found to be feeding on all types of blood meal. In addition, *C. brevatarsis* was also found to feed on both horse and dog blood in this study (Table 3; Figure 5).

## 4. Discussion

In this study, up to 12 *Culicoides* species were found in horse-raising areas, which were the AHS-positive and AHS-negative farms in Hua Hin district. A combination of microscopic and molecular identification of *Culicoides* species is useful and phylogenetic analysis has confirmed that each of the *Culicoides* species was closely related to the *Culicoides* sequences of Thailand, Japan, France, and Vietnam. The results of this study showed that *C. oxystoma* and *C. imicola* were the two most abundant *Culicoides* species in horse farms, which is similar to the previous study in India [14], but in contrast with the study of Choocherd [18]. In the study of Choocherd, the comparison of four different light traps for collecting *Culicoides* was conducted, and the ultraviolet fluorescent (UV-FLR) light trap was recommended [18]. They identified 26 *Culicoides* species in two horse farms in Hua Hin district, Prachuab Khiri Khan [18]. However, fewer *Culicoides* species were reported in this study, which may be due to the different time duration (8 months vs. 1 month), the seasonality of the species, the breeding sites, the availability of hosts, the habitat, etc. [2]. Moreover, the previous study identified *Culicoides* in all stages and genders, but this study only focused on fully engorged females [18]. *Culicoides oxystoma* has also previously been found near the beach in Chonburi province, Thailand [27]. The result of the present study is concordant with the previous study in Southern Thailand, India, and Senegal [3,14,28], which found more *C. oxystoma* than *C. imicola*.

The host preference of *Culicoides* spp. for blood meals can vary from mammals to poultry species due to the *Culicoides* species, host availability [29], geographical area, and use of target primers for blood meal identification [30]. The study of host preferences helps to identify the potential hosts that may be susceptible to AHSV. Several host preference studies have been conducted elsewhere, such as in Spain [31], Tunisia [32], Romania [33], Serbia [34], and Brazil [35], but there are few reports in Thailand [3,15,17]. One study that detected *Culicoides* blood meal collected from the Southern part of Thailand mentioned that biting midges may also act as potential vectors for leishmaniosis and trypanosomiasis [3]. Therefore, all hosts found as blood meal of *Culicoides* in this study may also be at risk of those parasite infections.

*Culicoides oxystoma* and *C. imicola* have been reported to feed on horse blood and are potential vectors in many countries [28]. *Culicoides oxystoma* preferred to feed on horses’ and other vertebrates’ blood [29] and has been marked as a potential vector of AHSV in Senegal [28]. In the current study, *C. oxystoma* was found to feed on a wide range of mammals as blood meal including horses, dogs, pigs, and humans, which is similar to the previous study in Senegal [28]. In addition, *C. imicola* was found to feed on horses, pigs, dogs, and human blood in this study. As similar to the previous study, *C. imicola* has also been reported to feed on human, goat, sheep, dog, and avian blood in Tunisia [30]. Moreover, *Culicoides* in Thailand has been reported to feed on chicken, cattle, and buffalo as the blood meal [15]. However, *C. oxystoma* has never been reported to feed on human blood in any other study in Thailand before. Therefore, the finding of the current study is the first report of human blood in *C. oxystoma*. Furthermore, *C. oxystoma* and *C. imicola* can be marked as potential vectors of AHSV in Thailand.

In 2020, the first AHS outbreak in Thailand served as a warning to quarantine imported live animals from the endemic areas. However, one of the preventive measures for AHS is to use insect nets, for instant “fly proof″, in combination with the AHS vaccination that was also recommended by the World Organization for Animal Health or Office International des Epizooties (OIE). However, non-strictly established insect nets are not efficient to protect horses from the biting of *Culicoides*. This is evident because we found more female *Culicoides* trapped at position A (within the insect nets) than at position B (outside of the insect nets). Previously, feeding patterns of biting midges (*C. pulicaris* and *C. obsoletus*) showed that they can feed on a range of vertebrate hosts but prefer cattle and other livestock in adjacent areas [36]. Therefore, more *Culicoides* were trapped close to the animal. There may be a distinct preference for the size of their host [36]. In our study, most of the horses were kept inside a horse barn with insect nets during the nighttime. This may be due to the hot and humid weather in Thailand, which is not suitable for using sealed insect nets because it is too hot and there is poor air ventilation in some areas. The owner has to open the insect net sometimes and the gate was not closed completely in every instance. The current study suggests all stakeholders should strongly focus on the prevention and control of emerging and zoonotic vector-borne diseases that may affect animals and human health in the future.

## 5. Conclusions

This study has identified the potential AHS *Culicoides* species in the Hua Hin district for the first time. At least 10 *Culicoides* species that feed on horses’ blood and may be the potential vector relating to AHS transmission were noted during the AHS outbreak in Thailand. Two *Culicoides* species (*C. oxystoma* and *C. imicola*) also prefer human blood as a blood meal. The detection and survey of possible zoonotic pathogens in human-suckling *Culicoides* can also be identified. Future attention on preventing AHS should be strongly and continuously conducted in the horse-raising industry.

## Figures and Tables

**Figure 1 insects-14-00369-f001:**
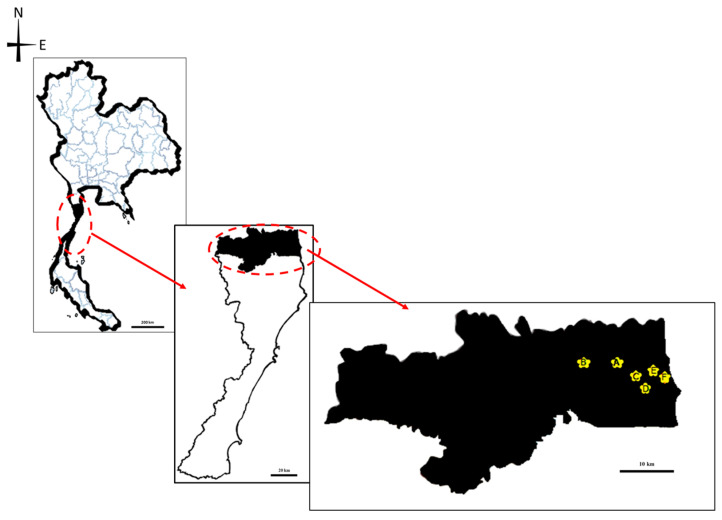
Map of the studied area in Hua Hin district, Prachuab Khiri Khan province, Thailand.

**Figure 2 insects-14-00369-f002:**
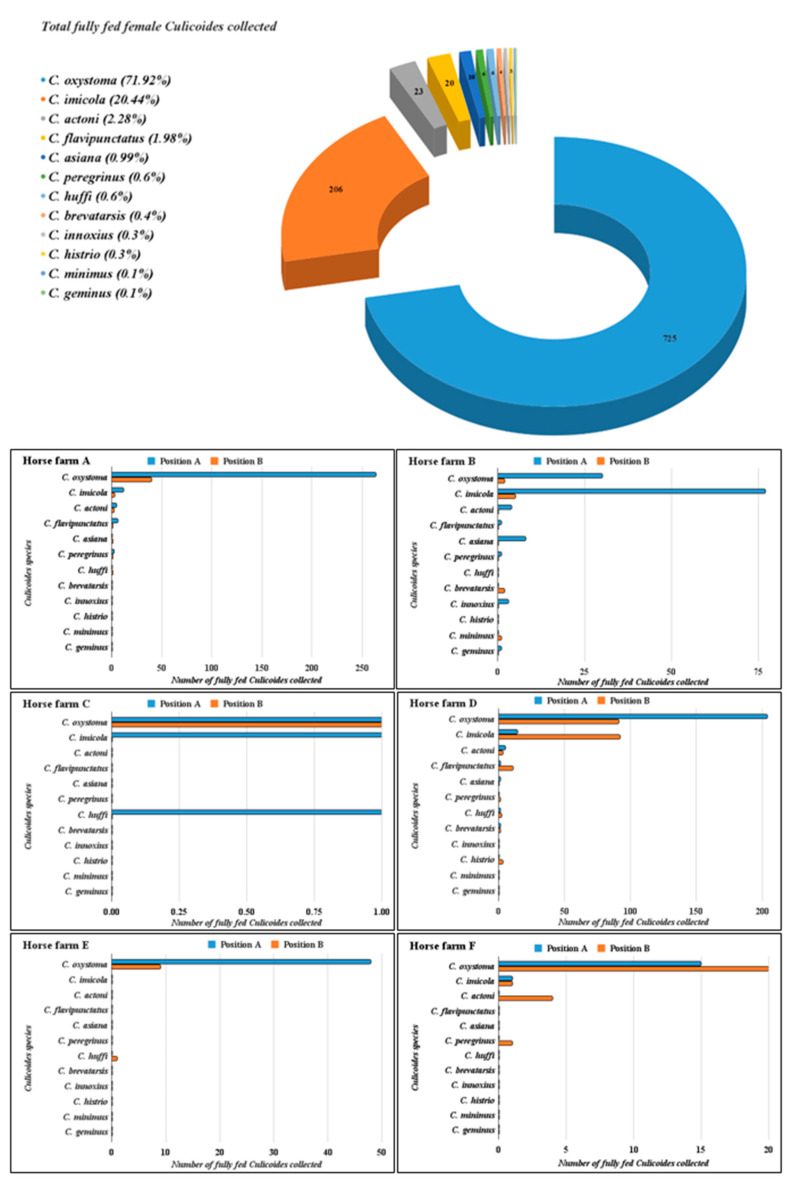
Number of *Culicoides* species collected from two positions at each horse farm in Hua Hin district, Prachuab Khiri Khan province, Thailand. Numbers in the pie chart are indicating numbers of each *Culicoides* species collected in this study.

**Figure 3 insects-14-00369-f003:**
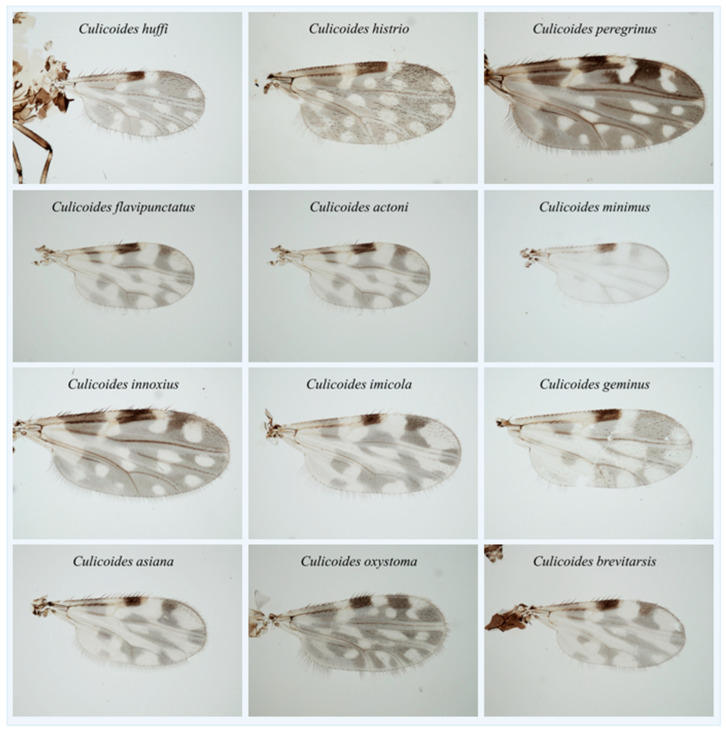
Wings of the 12 female *Culicoides* species collected in horse farms in Hua Hin district, Prachuab, Khiri Khan province, Thailand.

**Figure 4 insects-14-00369-f004:**
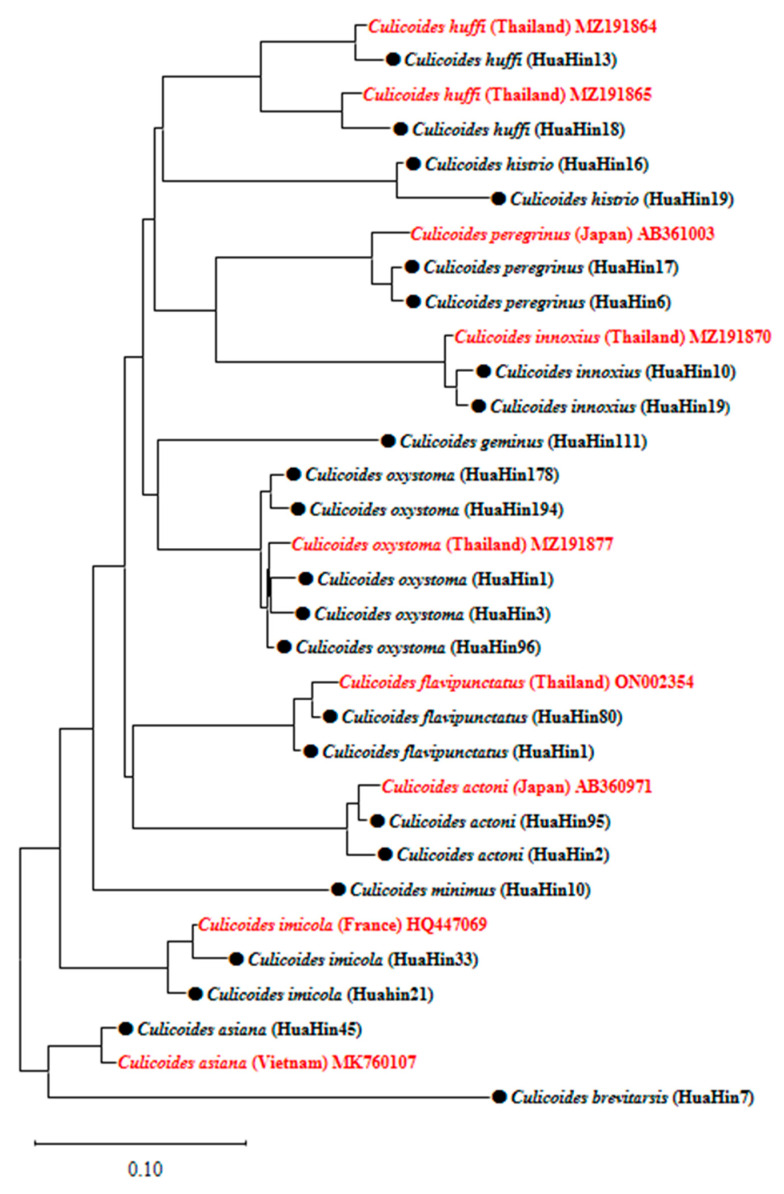
Phylogenetic tree of *Culicoides* collected in horse farms in Hua Hin district, Prachuab Khiri Khan province, Thailand. Reference sequences are marked in red.

**Figure 5 insects-14-00369-f005:**
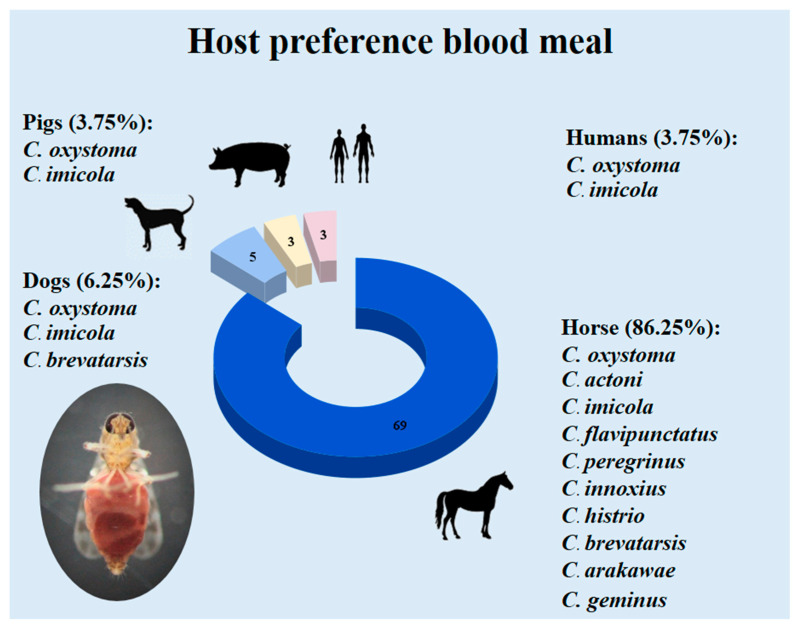
Host preference blood meal of the *Culicoides* species collected in horse farms in Hua Hin district, Prachuab Khiri Khan province, Thailand. Numbers in the pie chart are indicating number of *Culicoides* that feeding on each host blood meal.

**Table 1 insects-14-00369-t001:** Number of collected female *Culicoides* from six horse farms in Hua Hin district, Prachuab Khiri Khan Province, Thailand.

Horse Farm	Number of Collected Female *Culicoides*		
	Position A (Near the Horse <2 m) (%)	Position B (Far from the Horse >5 m) (%)	Total (%)
A	289 (85.5%)	49 (56%)	338 (33.5%)
B	125 (92.6%)	10 (4.7%)	135 (13.4%)
C	3 (75.0%)	1 (25.0%)	4 (0.40%)
D	227 (52.7%)	204 (47.3%)	431 (42.8%)
E	48 (82.8%)	10 (17.2%)	58 (5.8%)
F	16 (38.1%)	26 (61.9%)	42 (4.2%)
Total	708 (70.2%)	300 (29.8%)	1008 (100%)

**Table 2 insects-14-00369-t002:** Identification of *Culicoides* species in horse farms in Hua Hin district, Prachuab Khiri Khan province, Thailand.

*Culicoides* Species	Horse Farm A *		Horse Farm B *		Horse Farm C *		Horse Farm D *		Horse Farm E *		Horse Farm F		Total	(%)
	Position A	Position B	Position A	Position B	Position A	Position B	Position A	Position B	Position A	Position B	Position A	Position B		
*C. oxystoma*	264	40	30	2	1	1	204	91	48	9	15	20	725	(71.92%)
*C. imicola*	12	3	77	5	1	0	14	92	0	0	1	1	206	(20.44%)
*C. actoni*	5	2	4	0	0	0	5	3	0	0	0	4	23	(2.28%)
*C. flavipunctatus*	6	1	1	0	0	0	1	11	0	0	0	0	20	(1.98%)
*C. asiana*	0	1	8	0	0	0	1	0	0	0	0	0	10	(0.99%)
*C. peregrinus*	2	1	1	0	0	0	0	1	0	0	0	1	6	(0.60%)
*C. huffi*	0	1	0	0	1	0	1	2	0	1	0	0	6	(0.60%)
*C. brevatarsis*	0	0	0	2	0	0	1	1	0	0	0	0	4	(0.40%)
*C. innoxius*	0	0	3	0	0	0	0	0	0	0	0	0	3	(0.30%)
*C. histrio*	0	0	0	0	0	0	0	3	0	0	0	0	3	(0.30%)
*C. minimus*	0	0	0	1	0	0	0	0	0	0	0	0	1	(0.10%)
*C. geminus*	0	0	1	0	0	0	0	0	0	0	0	0	1	(0.10%)
Total	338		135		4		431		58		42		1008	(100.00%)

*: African horse sickness-positive farm.

**Table 3 insects-14-00369-t003:** Host preference blood meal of female *Culicoides* collected from AHS-positive horse farms in Hua Hin district, Prachuab Khiri Khan, Thailand, using PNOC gene.

*Culicoides* Species/Host Species	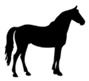 *Equus caballus*Position A/B (%)	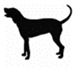 *Canis lupus familiaris*Position A/B (%)	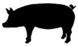 *Sus scrofa*Position A/B (%)	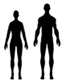 *Homo sapiens*Position A/B (%)
*C. oxystoma*	15/10	(36.23%)	2/1	(60.00%)	2/ND	(66.67%)	2/ND	(66.67%)
*C. peregrinus*	2/1	(4.35%)	ND		ND		ND	
*C. actoni*	7/7	(20.29%)	ND		ND		ND	
*C. flavipunctatus*	4/5	(13.04%)	ND		ND		ND	
*C. imicola*	7/3	(14.49%)	1/ND	(20.00%)	ND/1	(33.33%)	ND/1	(33.33%)
*C. brevatarsis*	1/ND	(1.45%)	ND/1	(20.00%)	ND		ND	
*C. innoxius*	3/ND	(4.35%)	ND		ND		ND	
*C. histrio*	ND/2	(2.90%)	ND		ND		ND	
*C. arakawae*	ND/1	(1.45%)	ND		ND		ND	
*C. geminus*	1/ND	(1.45%)	ND		ND		ND	
Total (80 females *Culicoides*)	40/29	(86.25%)	3/2	(6.25%)	2/1	(3.75%)	2/1	(3.75%)

ND = Not detected.

## Data Availability

The use of animals and collection protocol was approved by the Institutional Animal Care and Use Committee of Kasetsart University (approval number; ACKU64-VET-015).

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
