# Peer review of "Molecular Identification of Culicoides Species and Host Preference Blood Meal in the African Horse Sickness Outbreak-Affected Area in Hua Hin District, Prachuap Khiri Khan Province, Thailand"

_insects, 2023, doi:10.3390/insects14040369_

Round 1

Reviewer 1 Report

Insects: Molecular identification of Culicoides species and host preference blood meal in the African horse sickness outbreak-affected area in Hua Hin district, Prachuap Khiri Khan province, Thailand

The authors present a vector incrimination study for the area concerned in the 2020 AHS outbreak in Thailand. Culicoides host preference data for the area was scant prior to the outbreak and as such this study is important for the identification of species and their host preference to determine putative vectors of AHS. The study collected Culicoides from 5 AHS affected sites another non affected by AHS in 2020. Morphological and molecular techniques were used to identify Culicoides to species level and the host on which engorged females had fed. Catches were dominated by Culicoides oxystoma and C. imicola, both of which have been implicated as vectors of AHS and bluetongue virus throughout their extensive ranges and previously reported as dominant in Thailand. Both species were observed to have fed on horse hosts.

The manuscript would benefit from editing for English language use, particularly phrasing. The frequency of trapping at the field sites is not reported, nor time of year, factors that strongly influence the abundance and presence of Culicoides species. Only 26 specimens were sequenced for identification, and this, with the low trapping effort and few collected specimens means that the study cannot be considered to be a survey of the Culicoides taxa of this region of Thailand.

A greater number of Culicoides were collected within netted stables with horses, which is unsurprising due to the close association of Culicoides and livestock, but suggests the nets have no value in bite mitigation. The collection location of the 80 insects used for blood meal analysis was not disclosed, but if the engorged females were collected from close proximity to horses within a net then the ‘host preference’ of the majority of the Culicoides tested may an artefact of the collection method.

I have included further notes below.

Line 50: AHS is not endemic in Europe or Asia etc these are exotic outbreak regions.

Line 55 Spelling C. imicola

Line 107 and 120: DNA extraction method/kit used? What was the rationale for specimen selection for sequencing? Voucher specimens?

Line 234: does not really justify the use of PNOC over COI. Why would you want to avoid detection of old blood meals?

Line 248 and 253: repetition.

Line 250: Please rephrase, I do not understand this sentence.

Line 264: Was the net aperture small enough to restrict access by C. imicola, which is tiny? Is netting of stables practicable in this scenario?

Author Response

Response to reviewer’s comments

Reviewer 1

Q1.

Line 50: AHS is not endemic in Europe or Asia etc these are exotic outbreak regions.

Answer: Thank you for the reviewer’s comment. We have changed the sentence to ‘It is an arthropod-borne noncontagious disease of Equidae which is endemic in tropical and subtropical areas of Africa [1], epizootic in Europe and reported in the Middle East, India and Asia [2].’

Q2.

Line 55 Spelling C. imicola

Answer: Thank you the reviewer’s comment. We have spelled to ‘Culicoides (C.) imicola’.

Q3.

Line 107 and 120: DNA extraction method/kit used? What was the rationale for specimen selection for sequencing? Voucher specimens?

Answer: We apologize for not including the DNA extraction part in the MS. The DNA extraction were conducted after morphological examination with at least 20% of total specimens from each trap and each farm. We have added ‘DNA extraction: Culicoides abdomen containing host blood meal was individually preserved in absolute ethanol before DNA extraction. DNA extraction was conducted us-ing DNA extraction kit (MACHEREY-NAGEL, Germany). DNA concentration was meas-ured using spectrophotometer (Eppendorf, Germany) and stored in -20 C as DNA tem-plate.’ To the manuscript.  

Q4. Line 234: does not really justify the use of PNOC over COI. Why would you want to avoid detection of old blood meals?

Answer: Thank you for asking this question. The reason why we have selected PNOC over the COI was due to the mixed blood meal may interfered the sequencing result as mentioned in the previous study.

Q5. Line 248 and 253: repetition.

Answer: Thank you the reviewer’s comment. We have revised the discussion as marked in yellow.

Q6. Line 250: Please rephrase, I do not understand this sentence.

Answer: Thank you the reviewer’s comment. We have revised the discussion as marked in yellow.

Q7. Line 264: Was the net aperture small enough to restrict access by C. imicola, which is tiny? Is netting of stables practicable in this scenario?

Answer: Actually, we have not measure the mesh of insect nets that each farm used. However, netting of stables seem to be practicable in this scenario because there is no more AHS case reported. In addition, good air ventilation and controlling of temperature using electric fans is also needed.

Reviewer 2 Report

In this paper, 14 Culicoides species were identified near horse stables in Hua Hin district, Prachuab Khiri Khan province, Thailand, and statistical analysis was conducted. I think this manuscript is worth publishing after the following modifications:

1. It is recommended to move result 3.1 to the Materials and Methods section.

2. Please make histogram/pie chart for Table 2 to better represent the results.

3. Although I can understand, please reveal the meaning of the red font in Figure 3 in the legend.

4. Where is Figure 4?

5. How to select 80 samples from 1,008 female culicoides mosquitoes for host preference blood meal identification? Can deviations be avoided?

6.Page 10, line 209: Please revise "C. oxystoma and Culicoides imicola". In addition, please check all "Culicoides" that can be abbreviated as "C." in the entire manuscript.

Author Response

Reviewer 2  

Q1. It is recommended to move result 3.1 to the Materials and Methods section.

Answer: Thank you for the reviewer’s suggestion. We have moved result 3.1 to the MM section. Modification of the topic in result were also revised accordingly. We also have revised Figure 1 by removing pictures of trap and coordenates from Figure 1 as the third reviewer suggested.

Q2. Please make histogram/pie chart for Table 2 to better represent the results.

Answer: We appreciate for the reviewer’s comment. We have added a new Figure 2 based on the information of table 2 and re-arranged all figures afterward. 

Q3. Although I can understand, please reveal the meaning of the red font in Figure 3 in the legend.

Answer: Thank you for the reviewer’s suggestion.  We have added ‘Reference sequences are marked in red.’   in Figure 4 (Re-arranged Figure).

Q4. Where is Figure 4?

Answer: We apologize for the technical error, we have added Figure 5 in the revised manuscript.

Q5. How to select 80 samples from 1,008 female culicoides mosquitoes for host preference blood meal identification? Can deviations be avoided?

Answer: Thank you for the reviewer’s question. In fact, DNA extraction was extracted from 250 Culicoides specimens after microscopic examination. All Culicoides species trapped from each farm and each trap were selected. The proportion of selected specimens for molecular identification were fall within 20% of Culicoides from each trap, each farms. The confirmation of Culicoides species using COI gene were conducted in 26 DNA samples as it was used to confirm the microscopic identification. Indeed, blood meal identification was conducted in 149 DNA samples using PNOC gene. Then, there were 90 samples out of 149 samples in blood meal identification as clear positive band and only 80 sequences were confirmed by bioinformatics analysis. Therefore, we have not only picked 80 samples for molecular identification by chance but we have systematically selected samples to make sure that all Culicoides species from all farms were included. 

Q6.Page 10, line 209: Please revise "C. oxystoma and Culicoides imicola". In addition, please check all "Culicoides" that can be abbreviated as "C." in the entire manuscript.

Answer: Thank you for the reviewer’s comments. We apologize for errors. We have corrected accordingly.

Reviewer 3 Report

The authors identify Culicoides species and its blood-feeding habits after a AHSV outbreak in a district of Thailand. The studies related to vector-pathogen-hosts interactions are really interesting in therms of One Health issues. However, the MS needs to be improved in several aspects, especially the M&M section, detailed bellow. In addition, I strongky recommend to check the english with a native english researcher:

Simple summary

1. Line 14: remove the word "transmitted". It does not make sense here.

2. Line 16: replace "which" with "where"

3. The species C. imicola is misspelled in line 18 (C. immicola) and several times. Check it throughout the document.

4. Line 18: "Have consumed horses". Midges do not consume animals, they bite animals and suck blood.

5. Line 19: include the word "vector" after AHS and "and" after species.

Abstract

1. Line 22: "know as Culicoides" this is the genus name of the group so I suggest to change the sentence to: haematophagous insects from genus Culicoides

2. Line 24: "potential" what? you mean potential vector Culicoides species?

3. Line 25: remove "transmitted" before vectors. It is redundant.

4. Line 37.: why 24 of 26 DNA samples?

5. Line 38 again the use of "consume". Change it to "fed on". Check this word throughout the document.

6. Line 43-44: The sentence is confusing, please rephrase it.

Introduction

1. Line 52: Repace "which relates" with "related". Remove the word "the before midge

2. Please, include the references of the statements in line 54-55 and 55-56.

3. Line 56: replace " affected horsen farm" with "transmission".

4. Line 59: replace "the effective initialization" with " an effective begining"

5. Line 61: write "to" after "helps"

6. Line 72: replace "Leishmaniasis" with "leishmaniosis" and "relating" with "related".

7. Line 73: there is an extra "of". Please remove this word after Culicoides

8. Line 77: I suggest to move "in Thailand" to the begining of the sentence (In Thailand, Culicoides species have been....).

Materials and Methods

This part needs to be extensively

1. What does DLD means?

2. Lines 92-93: this is one of the objectives (evaluate the nets) and should be placed in the Introduccion section explaining the use of nets for Culicoides control

3. Lines 91-92: "purchased from" and the website I believe it is not necessary to be included.

4. Samples were collected alive or you fill the collectors with ethanol?

5. How often do you conduct the samples? One night per week? How many days?

6. The position A and B are not included in this section. Position A is inside and behind the net and position B outdoors? The distance between traps should be included

Results

1.Farm characteristics should be placed in M&M section including information about avalaiable hosts, type of farm, if is located in urban, rural, suburban or natural area.

2. Lines 138 and 139: I suggest to replace the sentence with " This study included five farms with positive history of AHS (farms A-E) and one farm that was historicaly negative to AHS (farm F)"

3. The pictures of each trap in figure 1 are irrelevant, I suggest to remove it. The coordenates could be placed in the text and keep the figure just with the map.

4. Line 146: I suggest to change the title of the subsection with just "3.2. Culicoides identified in horse farms" (Culicoides without italics since the other words are in italics).

5. Paragraph 150-152 should be improved in terms of use of english and structure. There is also too much information that the reader could just check it in table 1. 

6. The filogenetic analysis of figure 3 is poor. I will include more information in the results section and in the M&M and discussion.

7. Line 191: Culicoides brevitarsis could be abberviated as C. brevitarsis

8. In table 3 I would include the position A and B since it is interesting to explore the endofilic/endophagic or exophilic/exophagic behaviour of the different Culicoides species.

Discussion

1. The first paragraph should be placed in Introduction section

2. The information of line 206 does not make sense here.

3. Line 209: C. imicola must be abbreviated. 

4. Line 110: which previous studies? where?

5. Line 217. Include please the duration in Material and Methods section. In fact, the species collected could be also related to many other thinks like the seasonality of the species, the breeding sites, the avaliability of hosts, the habitat etc, etc, please, explore it!

6. Line 227: bluetongue virus? why? I believe is too late to talk about BTV since the MS is related to AHSV. If you decided to write about BTV, please include it in the introduction.

7. Line 232: leishmaniOsis

8. Line 234: this information is repeated before.

9. Lines 237-238 should be placed in M&M section.

10. Paragraph 234-258 is quite confusing and the information needs to be better structured for better understanding. 

11. Line 255: C. oxystoma must be written with full genus name after the dot.

12. Discuss the phylogeny and the feeding behaviour

13. Line 275: I do not understand the meening of this sentence. You mean that you want to check for vertical transmission? detect AHSV in subadult stages??

Author Response

Reviewer 3

Simple summary

Q1. Line 14: remove the word "transmitted". It does not make sense here.

Answer: We appreciate the reviewer’s suggestion. We have remove ‘transmitted’ from the sentence. 

Q2. Line 16: replace "which" with "where"

Answer: We appreciate the reviewer’s suggestion. We have changed ‘which’ to ‘where’ as suggested.

Q3. The species C. imicola is misspelled in line 18 (C. immicola) and several times. Check it throughout the document.

Answer: We appreciate for the reviewer’s suggestion. We have carefully corrected to ‘C. imicola’ as commented.

Q4. Line 18: "Have consumed horses". Midges do not consume animals, they bite animals and suck blood.

Answer: Thank you for pointing out the mistake. We have used ‘feeding on’, ‘fed on’ instead of ‘consume’ as suggested accordingly.  

Q5. Line 19: include the word "vector" after AHS and "and" after species.

Answer: We appreciate the reviewer’s suggestion. We have added ‘vector’ and ‘and’ as suggested.

Abstract

Q1. Line 22: "know as Culicoides" this is the genus name of the group so I suggest to change the sentence to: haematophagous insects from genus Culicoides

Answer: We appreciate the reviewer’s suggestion. We have revised as commented.

Q2. Line 24: "potential" what? you mean potential vector Culicoides species?

Answer: It is mean possible of Culicoides species that can transmit AHSV.  

Q3. Line 25: remove "transmitted" before vectors. It is redundant.

Answer: We appreciate the reviewer’s suggestion. We have removed it as suggested.

Q4. Line 37.: why 24 of 26 DNA samples?

Answer: Out of 26 DNA samples were used for identification of Culicoides species by PCR. However, due to the sequences quality so, sequencing analysis can be confirmed in 23 sequences. We have revised to ‘23 DNA samples’.

Q5. Line 38 again the use of "consume". Change it to "fed on". Check this word throughout the document.

Answer: We appreciate the reviewer’s suggestion. We have revised accordingly.

Q6. Line 43-44: The sentence is confusing, please rephrase it.

Answer: We appreciate the reviewer’s suggestion. We have revised to ‘This study revealed the species of Culicoides in Hua Hin district, Thailand after the AHS outbreak.’.

Introduction

Q1. Line 52: Repace "which relates" with "related". Remove the word "the before midge

Answer: We appreciate the reviewer’s suggestion. We have revised accordingly.

Q2. Please, include the references of the statements in line 54-55 and 55-56.

Answer: We appreciate the reviewer’s suggestion. We have added a reference as recommended.

Q3. Line 56: replace " affected horsen farm" with "transmission".

Answer: We appreciate the reviewer’s suggestion. We have revised accordingly.

Q4. Line 59: replace "the effective initialization" with " an effective begining"

Answer: We appreciate the reviewer’s suggestion. We have revised accordingly.

Q5. Line 61: write "to" after "helps"

Answer: We have revised as recommended.

Q6. Line 72: replace "Leishmaniasis" with "leishmaniosis" and "relating" with "related".

Answer: We have revised as recommended.

Q7. Line 73: there is an extra "of". Please remove this word after Culicoides

Answer: We have revised to ‘In Thailand, Culicoides species of fauna in Thailand have been investigated since 1938, and ~100 Culicoides species were recently recorded’.

Q8. Line 77: I suggest to move "in Thailand" to the begining of the sentence (In Thailand, Culicoides species have been....).

Answer: We have revised to ‘In Thailand, Culicoides species of fauna in Thailand have been in-vestigated since 1938, and ~100 Culicoides species were recently recorded’.

Materials and Methods

This part needs to be extensively

Q1. What does DLD means?

Answer: we have added ‘Department of Livestock Development’ before ‘DLD’.

Q2. Lines 92-93: this is one of the objectives (evaluate the nets) and should be placed in the Introduccion section explaining the use of nets for Culicoides control

Answer: In fact, we have not aimed to evaluate the insect nets in this study. We have only try to get Culicoides samples as much as possible, so that we put the trap inside the insect net to be near the horse and outside of the insect net as it is far from the horse. This is to have an idea where to get more Culicoides and unexpectedly there were more Culicoides when trapped near the horse which was inside the insect net. To evaluate the effective of the insect net should be design the new study in the future. 

Q3. Lines 91-92: "purchased from" and the website I believe it is not necessary to be included.

Answer: We have removed website as commented.

Q4. Samples were collected alive or you fill the collectors with ethanol?

Answer: Samples were not alive. We have collection bottle filled with absolute ethanol as already referred to reference.

Q5. How often do you conduct the samples? One night per week? How many days?

Answer: Sampling was conducted once for 12 h from 6 PM to 6 AM as indicated.

Q6. The position A and B are not included in this section. Position A is inside and behind the net and position B outdoors? The distance between traps should be included

Answer: We have added ‘position A’ after ‘inside’ and ‘position B’ after ‘outside’. However, the distance between the trap was not able to add because the size of horse barn, different environment, and distance from the light were vary in each farm.

Results

Q1.Farm characteristics should be placed in M&M section including information about avalaiable hosts, type of farm, if is located in urban, rural, suburban or natural area.

Answer: We have moved result 3.1 to MM. We have added ‘All farms were located in suburban areas of Hua Hin district.’ in to this part. The Available hosts already mentioned here.

Q2. Lines 138 and 139: I suggest to replace the sentence with " This study included five farms with positive history of AHS (farms A-E) and one farm that was historicaly negative to AHS (farm F)"

Answer: As we have move to MM part. To avoid repeating the same sentence. We have revised ‘Five farms with positive history of AHS (farms A-E) and one farm that was historically negative to AHS (farm F) were included (Figure. 1.)’.

Q3. The pictures of each trap in figure 1 are irrelevant, I suggest to remove it. The coordenates could be placed in the text and keep the figure just with the map.

Answer: we have removed the picture of each trap and the coordenates from figure 1 as recommended. The coordenates were indicated in table 1. We have removed ‘(These images appearing personal identity of authors were used with the rights of publicity and privacy permission)’ from figure 1 caption as there is no images appearing personal identity.

Q4. Line 146: I suggest to change the title of the subsection with just "3.2. Culicoides identified in horse farms" (Culicoides without italics since the other words are in italics).

Answer: We have revised as recommended. It was re-arranged to 3.1.

Q5. Paragraph 150-152 should be improved in terms of use of english and structure. There is also too much information that the reader could just check it in table 1.

Answer: We have revised as recommended.

Q6. The filogenetic analysis of figure 3 is poor. I will include more information in the results section and in the M&M and discussion.

Answer: We appreciate the reviewer’s comment.

Q7. Line 191: Culicoides brevitarsis could be abberviated as C. brevitarsis

Answer: We have revised as suggested.

Q8. In table 3 I would include the position A and B since it is interesting to explore the endofilic/endophagic or exophilic/exophagic behaviour of the different Culicoides species.

Answer: We appreciate the reviewer’s comment. That would be very informative and useful. We have revised as suggested.

Discussion

Q1. The first paragraph should be placed in Introduction section

Answer: We have revised as recommended.

Q2. The information of line 206 does not make sense here.

Answer: We have removed the sentence.

Q3. Line 209: C. imicola must be abbreviated.

Answer: We have revised as suggested.

Q4. Line 210: which previous studies? where?

Answer: It was the collection of Culicoides midges near a buffalo shed in India.  We have added ‘in india’.

Q5. Line 217. Include please the duration in Material and Methods section. In fact, the species collected could be also related to many other thinks like the seasonality of the species, the breeding sites, the avaliability of hosts, the habitat etc, etc, please, explore it!

Answer: We have revised to ‘However, less Culicoides species were reported in this study, which may be due to the different time duration (8 months vs 1 month), the seasonality of the species, the breeding sites, the availability of hosts, the habitat and etc. [2].’

Q6. Line 227: bluetongue virus? why? I believe is too late to talk about BTV since the MS is related to AHSV. If you decided to write about BTV, please include it in the introduction.

Answer: We have removed ‘and bluetongue virus’ from the sentence.

Q7. Line 232: leishmaniOsis

Answer: We have revised as suggested.

Q8. Line 234: this information is repeated before.

Answer: We have revised as suggested.

Q9. Lines 237-238 should be placed in M&M section.

Answer: We have moved and revised as suggested.

Q10. Paragraph 234-258 is quite confusing and the information needs to be better structured for better understanding.

Answer: We have revised as suggested.

Q11. Line 255: C. oxystoma must be written with full genus name after the dot.

Answer: We have revised as suggested.

Q12. Discuss the phylogeny and the feeding behaviour

Answer: We appreciate the reviewer’s comment. We have added a sentence ‘Combination of microscopic and molecular identification of Culicoides species is useful, phylogenetic analysis has confirmed that each of Culicoides species were closely related to the Culicoides sequences of Thailand, Japan, France and Vietnam.’ as recommended.

Q13. Line 275: I do not understand the meening of this sentence. You mean that you want to check for vertical transmission? detect AHSV in subadult stages?

Answer: We have removed the sentence.  

Round 2

Reviewer 1 Report

The authors have improved the manuscript significantly.

Author Response

Responses to the reviewer’s comments

Reviewer 1

The authors have improved the manuscript significantly.

Answer: Thank you very much. We appreciated the reviewer’s comments and suggestions. That’s helping to improving the manuscript a lot. 

Reviewer 3

Dear authors. Thanks for the revision of the manuscript. Here some minor comments

Answer: Thank you very much. We appreciated the reviewer’s comments and suggestions. That’s helping to improving the manuscript a lot.  We have checked typing errors, spelled in full for some abbreviations, and removed some repeating sentences in the manuscript.

Q1. The coordenates of the sampling sites must be included in M&M

Answer: Thank you for the reviewer’s suggestion. We have removed the coordenates from table 1 and  added ‘The location of each farm included farm A: 12°33'17.3"N 99°56'46.2"E, farm B: 12°34'34.4"N 99°53'09.9"E, farm C: 12°33'08.8"N 99°56'59.3"E, farm D: 12.546426"N, 99.955316"E, farm E: 12°33'12.3"N 99°57'26.7"E, and farm F: 12°33'00.6"N 99°57'35.6"E, respectively.’  in M&M.

Q2. Please, include the scale in figure 1

Answer: Thank you for the reviewer’s suggestion. We have added scales in figure 1.

Q3. The resolution of figure 2 needs to be improved.

Answer: We apologize for not carefully check the resolution of the figure 2. We have rebuilt figure 2 in a better resolution.

Q4. The word Culicoides should be typed as Culicoides since the title is in italics

Answer: We apologize for not change as suggested. We understood that a term ‘Culicoides’ is belong to the genus, and the genus is normally be written in italic.   Please guide us if we are not correct.

Reviewer 3 Report

Dear authors. Thanks for the revision of the manuscript. Here some minor comments:

1. The coordenates of the sampling sites must be included in M&M

2. Please, include the scale in figure 1

3. The resolution of figure 2 needs to be improved.

4. The word Culicoides should be typed as Culicoides since the title is in italics

Regards

Author Response

(The authors gave the same response as above.)
